# ER-to-lysosome-associated degradation acts as failsafe mechanism upon ERAD dysfunction

Elisa Fasana[1,3], Ilaria Fregno [1,3], Carmela Galli[1,3], Tatiana Soldà[1] & Maurizio Molinari [1,2]✉

## Abstract

The endoplasmic reticulum (ER) produces proteins destined to organelles of the endocytic and secretory pathways, the plasma membrane, and the extracellular space. While native proteins are transported to their intra- or extracellular site of activity, folding-defective polypeptides are retro-translocated across the ER membrane into the cytoplasm, poly-ubiquitylated and degraded by 26 S proteasomes in a process called ER-associated degradation (ERAD). Large misfolded polypeptides, such as polymers of alpha1 antitrypsin Z (ATZ) or mutant procollagens, fail to be dislocated across the ER membrane and instead enter ER-to-lysosome-associated degradation (ERLAD) pathways. Here, we show that pharmacological or genetic inhibition of ERAD components, such as the α1,2-mannosidase EDEM1 or the OS9 ERAD lectins triggers the delivery of the canonical ERAD clients Null Hong Kong (NHK) and BACE457Δ to degradative endolysosomes under control of the ER-phagy receptor FAM134B and the LC3 lipidation machinery. Our results reveal that ERAD dysfunction is compensated by the activation of FAM134B-driven ERLAD pathways that ensure efficient lysosomal clearance of orphan ERAD clients.

**Keywords** Endoplasmic Reticulum (ER); ER-associated Degradation (ERAD); ER-to-Lysosome-associated Degradation (ERLAD); ER-phagy; Protein Quality Control
**Subject Categories** Autophagy & Cell Death; Organelles; Post-translational Modifications & Proteolysis

## Introduction

Maintenance of cellular homeostasis relies on efficient clearance of folding-defective gene products. ERAD, for ER-associated protein degradation, is an acronym coined to describe the proteasomal clearance from the ER of misfolded pro-alpha factor in a reconstituted yeast system (McCracken and Brodsky, 1996). ERAD now defines all the pathways in eukaryotic cells that ensure recognition of folding-defective polypeptides in the ER lumen or membrane and control their transport into the cytosol for poly-ubiquitylation and degradation by 26 S proteasomes (Fig. 1A) (Christianson et al, 2023; Hebert et al, 2010; Sun and Brodsky, 2019).

Recently, it has been observed that large misfolded proteins fail to be transported across the ER membrane into the cytosol and fail therefore to enter ERAD pathways. These polypeptides are segregated in specialized ER subdomains and are eventually delivered to degradative endolysosomes or autolysosomes (as defined in (Bright et al, 2016; Huotari and Helenius, 2011) and (Klionsky et al, 2014), respectively, Fig. 1B) in Metazoan cells, or to vacuoles in yeast and plant cells (Rudinskiy and Molinari, 2023). The term ERLAD for ER-to-lysosome-associated degradation was coined to describe the delivery from the ER to degradative RAB7/LAMP1-positive endolysosomes of disease-causing ATZ polymers (Fregno et al, 2018; Fregno et al, 2021) or of misfolded procollagen (Forrester et al, 2019) (Fig. 1B, arrows 1 and 2, respectively). Nowadays, ERLAD is an umbrella term that embraces all the autophagic and non-autophagic pathways that deliver ERAD-resistant misfolded polypeptides from the ER to degradative endolysosomes/autolysosomes/vacuoles (Fig. 1B) (Fregno and Molinari, 2019; Gubas and Dikic, 2022; Klionsky et al, 2021; Rudinskiy and Molinari, 2023). Mechanistically, the clearance from the ER of ATZ polymers and of procollagen molecules relies on cycles of de-/re-glucosylation of N-linked oligosaccharides that prolong calnexin (CNX) binding to favor engagement of the ER membrane protein FAM134B (Fregno et al, 2018, 2021). FAM134B has been first described as a LC3-binding protein, whose role in autophagic clearance of ER subdomains is activated upon nutrient restriction (Khaminets et al, 2015). More recently, our group established that its role as an ER-phagy receptor (i.e., a protein that controls lysosomal clearance of ER portions and content) is also activated by the luminal accumulation of ERAD-resistant misfolded polypeptides (Forrester et al, 2019; Fregno et al, 2018, 2021). Disposal of ATZ monomers proceeds via ERAD. In contrast, clearance from cells of ATZ polymers relies on their delivery into lysosomal degradative compartments and involves ATG5 and ATG7 autophagy gene products that control LC3 lipidation (Chu et al, 2014; Fregno et al, 2018; Hidvegi et al, 2010; Kroeger et al, 2009; Pastore et al, 2013; Sun et al, 2023; Teckman and Perlmutter, 2000). Despite the intervention of the LC3 lipidation machinery, in all mammalian cell lines tested in our lab, the lysosomal clearance of ATZ polymers does not require the intervention of autophagosomes. Rather, it relies on SNARE proteins-driven fusion of ER

[1]Faculty of Biomedical Sciences, Institute for Research in Biomedicine, Università della Svizzera italiana (USI), 6500 Bellinzona, Switzerland. [2]School of Life Sciences, École Polytechnique Fédérale de Lausanne, 1015 Lausanne, Switzerland. [3]These authors contributed equally: Elisa Fasana, Ilaria Fregno, Carmela Galli.
✉E-mail: maurizio.molinari@irb.usi.ch

## A  ERAD pathways

## B  ERLAD pathways

**Figure 1.  Proteasomal and lysosomal pathways for clearance of misfolded proteins from the ER.**

(A) ER-associated degradation (ERAD) pathways rely on mannose processing by EDEM proteins that interrupts futile folding cycles in the CNX chaperone system and engages mannose-binding lectins (OS9). Retro-translocation machineries (Dislocons) built around membrane-embedded E3 ubiquitin ligases ensure client-specific dislocation across the ER membrane and poly-ubiquitylation that precedes proteasomal degradation. (B) ER-to-lysosome-associated degradation (ERLAD) pathways. For ATZ and procollagen, selection for ERLAD relies on cycles of de-/re-glucosylation that prolong CNX binding and lead to association with the FAM134B (Forrester et al, 2019; Fregno et al, 2018, 2021). ERLAD clients are eventually cleared in endolysosomes or in autolysosomes, where they are delivered via mechanistically distinct pathways shown with arrows 1–3.

subdomains or vesicles containing ATZ polymers with RAB7/LAMP1-positive endolysosomes (Fig. 1B, arrow 1) (Fregno et al, 2018, 2021). In contrast, autophagosomes are involved in disposal of procollagen molecules, which proceeds via macro-ER-phagy (Fig. 1B, arrow 2) (Forrester et al, 2019). A variety of ERLAD clients is found in the literature, but for few of them the degradative mechanisms have been characterized in molecular detail (Rudinskiy and Molinari, 2023). Certainly, a variety of ERLAD pathways does operate in eukaryotic cells. These catabolic programs are regulated by client- and tissue-specific ER-phagy receptors, whose activity in promoting clearance of ER subdomains is activated by the intraluminal or membrane accumulation of misfolded polypeptides, including FAM134B (Forrester et al, 2019; Fregno et al, 2018; Fregno et al, 2021), CCPG1 (Ishii et al, 2023; Smith et al, 2018) and FAM134B-2 (Kohno et al, 2019). Their activation ensures delivery of ER content to be removed from cells to degradative organelles via LC3-dependent delivery (Fig. 1B, arrow 1), macro-ER-phagy (arrow 2), or micro-ER-phagy (arrow 3).

Notably, ERAD inhibition delays, rather than blocks, degradation of ERAD clients, hinting at alternative pathways intervening to ensure efficient clearance of misfolded proteins generated in the ER under conditions of ERAD impairment or overload (Molinari, 2007). To uncover a possible interplay of proteasomal (ERAD) and lysosomal (ERLAD) quality control of defective gene products synthesized in the ER, we assessed the capacity of the ERLAD machinery to support clearance of Null Hong Kong (NHK) and BACE457Δ, two classical ERAD clients, upon pharmacologic or genetic ERAD inactivation. The NHK variant of alpha1 antitrypsin is a disease-causing folding-defective glycoprotein (Sifers et al, 1988). BACE457Δ is a folding-defective splice variant of beta-secretase (Molinari et al, 2002). Characterization of the machineries ensuring proteasomal clearance of NHK and BACE457Δ from the mammalian ER revealed general principles of ERAD: (i) the role of mannose processing by EDEM proteins to extract terminally misfolded polypeptides from the CNX folding cycle (Chiritoiu et al, 2020; Hirao et al, 2006; Liu et al, 1997, 1999; Molinari et al, 2003; Oda et al, 2003; Olivari et al, 2006); (ii) the engagement of OS9 ERAD lectins that deliver misfolded polypeptides to various client-specific dislocons embedded in the ER membrane that control ERAD clients retro-translocation into the cytosol for poly-ubiquitylation and proteasomal degradation (Fig. 1A) (Bernasconi et al, 2010a; Bernasconi et al, 2008; Christianson et al, 2008; Ninagawa et al, 2011; Sugimoto et al, 2017)); (iii) the involvement of members of the protein disulfide isomerase (Guerra et al, 2018; Molinari et al, 2002; Ushioda et al, 2008) and of the peptidyl-prolyl cis/trans isomerase superfamilies (Bernasconi et al, 2010b). As such, NHK and BACE457Δ are among the best-characterized clients of the ERAD machinery. Here, we monitor their fate in cells with pharmacologically- or genetically induced dysfunction of ERAD. Our data reveal the intervention of FAM134B-driven ERLAD pathways to warrant efficient clearance of misfolded ERAD clients in cells characterized by dysfunctional ERAD.

## Results

### BafA1 stabilizes NHK upon prolonged pharmacologic inhibition of 26S-proteasomes or α1,2- mannosidases

To assess a possible intervention of ERLAD to compensate for dysfunctional ERAD, the degradation of $^{35}$S-methionine/cysteine radiolabeled NHK was determined in mock-treated HEK293 cells (Fig. 2A, lanes 1–3 (for one representative experiment), B,C, mock (for quantification of three biological replicates)), in HEK293 cells incubated with the dipeptide boronic acid Bortezomib/PS341, a selective inhibitor of 26 S proteasomes (Adams et al, 1999) (Fig. 2A, lanes 4–5, 2B, PS), or with the alkaloid kifunensine (KIF), a selective inhibitor of α1,2-mannosidases of the glycosyl hydrolase 47 family members, which includes the three EDEM proteins (Elbein et al, 1990; Liu et al, 1999; Moremen and Molinari, 2006; Olivari and Molinari, 2007; Vallee et al, 2000) (Fig. 2A, lanes 8–9, 2C, KIF). The immunoisolation from cell lysates of radiolabeled NHK after 75 or 120 min of chase (Fig. 2A) reveals that both PS341 (Fig. 2B, PS) and KIF (Fig. 2C, KIF) delay, rather than block ERAD of NHK. The slower decay of NHK in cells with dysfunctional ERAD (i.e., in cells exposed to PS341 or to KIF) is further reduced upon lysosomal inactivation with BafA1 (Fig. 2B, lanes 6, 7, and 10,

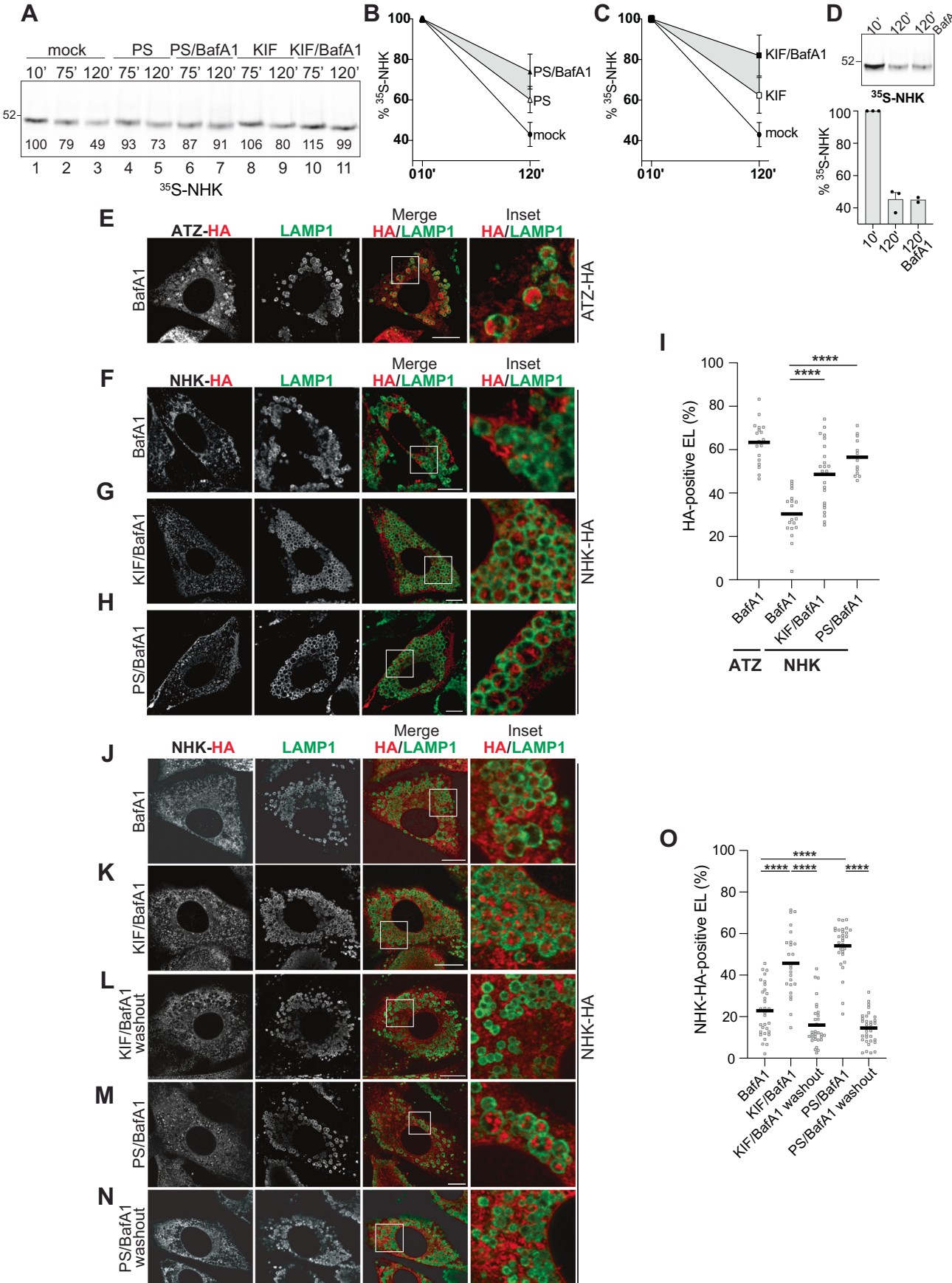

**Figure 2.  The ERAD client NHK is re-directed to LAMP1-positive endolysosomes for degradation upon pharmacologic inactivation of ERAD.**

(A) HEK239 cells transiently expressing the $^{35}$S-NHK are chased for the indicated time without (lanes 1–3) or with PS341 (lanes 4, 5), PS341/BafA1 (lanes 6, 7), KIF (lanes 8, 9), or KIF/BafA1 (10–11). Radiolabeled NHK is immunoisolated at the end of the indicated chase time and separated in SDS-PAGE. Densitometric quantification of the $^{35}$S-NHK band in this representative experiment is shown. (B) Quantification of $^{35}$S-NHK after a 10- or 120-min chase in cells mock-treated or exposed to PS341 or PS341/ BafA1. Individual data points at 120-min (%, normalized to 10′ mock): mock= 49, 37, and 49%; PS341 = 73, 55, and 52%; PS341/BafA1 = 91, 63, and 68%. (C) Same as (B) for Kif, KIF/BafA1. Individual data points at 120-min (%, normalized to 10-min mock): mock= 49, 37, and 49%; KIF = 80, 53 and 54%; KIF/BafA1 = 99, 64, and 83%. (D) BafA1 does not inhibit clearance of the ERAD client NHK. $n = 3$ for 10- and 120-min; $n = 2$ for 120-min BafA1. (E) ATZ, a canonical ERLAD client accumulates within LAMP1-positive endolysosomes in 3T3 cells exposed to BafA1. (F) The ERAD client NHK is not delivered within LAMP1-positive endolysosomes. (G, H) NHK is delivered and accumulates within endolysosomes inactivated with BafA1 upon ERAD inhibition with KIF (G) or with PS341 (H). (I) LysoQuant quantification of ATZ delivery within LAMP1-positive endolysosomes in (E), and of NHK in (F–H) ($n = 18, 19, 23$, and 13 cells, respectively). (J–N) Analysis of NHK delivery to LAMP1-positive endolysosomes as in (F–H) but with drug incubation of 8 h and with additional BafA1 washout conditions (L, N). (O) LysoQuant quantification of NHK accumulation within LAMP1-positive endolysosomes (J–N, $n = 31, 24, 31, 31$, and 33 cells respectively). Data Information: (B, C) $N = 3$ independent experiments, mean +/− SEM is shown. (D) Mean +/− SEM is shown. (I, O) Mean is shown. $N = 3$ independent experiments. One-way analysis of variance (ANOVA) and Dunnett's multiple comparison test, ****$P < 0.0001$. (E–H, J–N) Scale bars: 10 µM, Insets are shown with white squares. Source data are available online for this figure.

11 and the gray zone in Fig. 2B,C). Notably, lysosomal inhibitors per se, have no evident inhibitory activity on NHK disposal (Liu et al, 1999) (Fig. 2D). Thus, lysosomal inactivation does not impact on clearance of NHK (Fig. 2D), unless the ERAD pathway is dysfunctional (Fig. 2A, lanes 6, 7, 10, 11 and gray zones in Fig. 2B,C).

## The ERAD client NHK is normally not delivered to endolysosomes for clearance

The canonical ERLAD client ATZ is delivered to LAMP1-positive endolysosomes for clearance (Fregno et al, 2018, 2021). Consistently, when mouse 3T3 cells expressing HA-tagged ATZ were incubated with BafA1 to inhibit lysosomal hydrolases and preserve delivered material in the lumen of degradative organelles (Klionsky et al, 2008), ATZ accumulates in endolysosomes that display LAMP1 at the limiting membrane as seen in Confocal laser scanning microscopy (CLSM) (Fig. 2E). ATZ delivery to the degradative compartment is quantified with LysoQuant (Morone et al, 2020), a deep-learning-based analysis software for segmentation and classification of fluorescence images (Fig. 2I, ATZ+BafA1). The ERAD client NHK, whose clearance from cells is not delayed upon inhibition of lysosomal enzymes with BafA1 (Fig. 2D) (Liu et al, 1999) is poorly delivered within LAMP1-positive degradative compartments (Fig. 2F,J and quantification in Fig. 2I,O, NHK+BafA1).

## Pharmacologic inhibition of ERAD triggers lysosomal delivery of NHK

Motivated by the results of the biochemical analyses showing that inactivation of lysosomal hydrolases stabilizes NHK when ERAD is inactive (Fig. 2A–C), we examined the delivery to endolysosomes of NHK upon pharmacologic inhibition of the ERAD machinery in mouse 3T3 cells. CLSM reveals the enhanced accumulation of NHK in endolysosomes that display LAMP1 at the limiting membrane in cells where ERAD has been inhibited with KIF (Fig. 2G,I), or with PS341 (Fig. 2H,I). Thus, inhibition of ERAD at early (KIF) and late steps (PS341) of client selection, activates channeling of misfolded polypeptides into ERLAD pathways that deliver them to degradative LAMP1-positive compartments. The degradative nature of the LAMP1-positive endolysosomes where NHK accumulates in cells treated with BafA1 (Fig. 2F–I,K,M,O) is confirmed by the disappearance of the NHK immunoreactivity from the endolysosomal lumen upon BafA1 washout (Fig. 2L,N,O) and (Fregno et al, 2018).

## Genetic inhibition of ERAD upon silencing of EDEM1 triggers delivery of NHK to degradative endolysosomes

EDEM1 is an active α1,2-mannosidase that ensures the extraction of terminally misfolded glycoproteins from the CNX chaperone system to direct them for proteasomal degradation (Molinari et al, 2003; Oda et al, 2003; Olivari et al, 2006; Olivari and Molinari, 2007). Silencing of EDEM1 expression delays ERAD of glycoproteins (Molinari et al, 2003). To verify whether silencing of EDEM1 expression diverts NHK into the ERLAD pathway to compensate for ERAD dysfunction, we compared delivery of HA-tagged NHK in the LAMP1-positive endolysosomes in HEK293 cells expressing a scrambled short hairpin RNA (shCTRL), or a short hairpin RNA targeting the EDEM1 sequence (shEDEM, Fig. 3A). Cells were exposed to BafA1 to preserve the NHK fraction possibly delivered in the lumen of endolysosomes. CLSM reveals the enhanced accumulation of NHK in endolysosomes that display LAMP1 at the limiting membrane in cells with reduced EDEM1 levels (Fig. 3B, lower panels), which has been quantified by LysoQuant (Fig. 3C).

## Genetic inhibition of ERAD upon silencing of OS9 proteins triggers delivery of NHK to degradative endolysosomes

De-mannosylation of N-glycans on misfolded glycoproteins generates binding sites for the OS9 ERAD lectins, which are expressed in two ER stress-induced splice variants in the ER lumen (Bernasconi et al, 2008). OS9 association facilitates proteasomal clearance of NHK by promoting the delivery of the misfolded polypeptide to the HRD1/SEL1L retro-translocation machinery (Fig. 1A) (Bernasconi et al, 2010a; Bernasconi et al, 2008; Christianson et al, 2008). We and others previously showed that silencing of OS9 expression delays ERAD of NHK (Bernasconi et al, 2008; Christianson et al, 2008). To verify whether under these conditions NHK was diverted into the ERLAD pathways to compensate ERAD dysfunction, we monitored NHK delivery in the LAMP1-positive endolysosomes in HEK293 cells expressing a control short hairpin scrambled RNA (shCTRL), or two short hairpin RNAs targeting the OS9.1 and OS9.2 sequences (shOS9.1/ OS9.2A and shOS9.1/OS9.2 B, Fig. 3D). shOS9.1/OS9.2A reduced by about 30% the cellular level of OS9.1, and by about 70% the level of OS9.2 (Fig. 3D). shOS9.1/OS9.2 B significantly reduced only the level of OS9.2 (Fig. 3D). CLSM reveals that in cells expressing shOS9.1/OS9.2A NHK delivery to the degradative LAMP1-positive

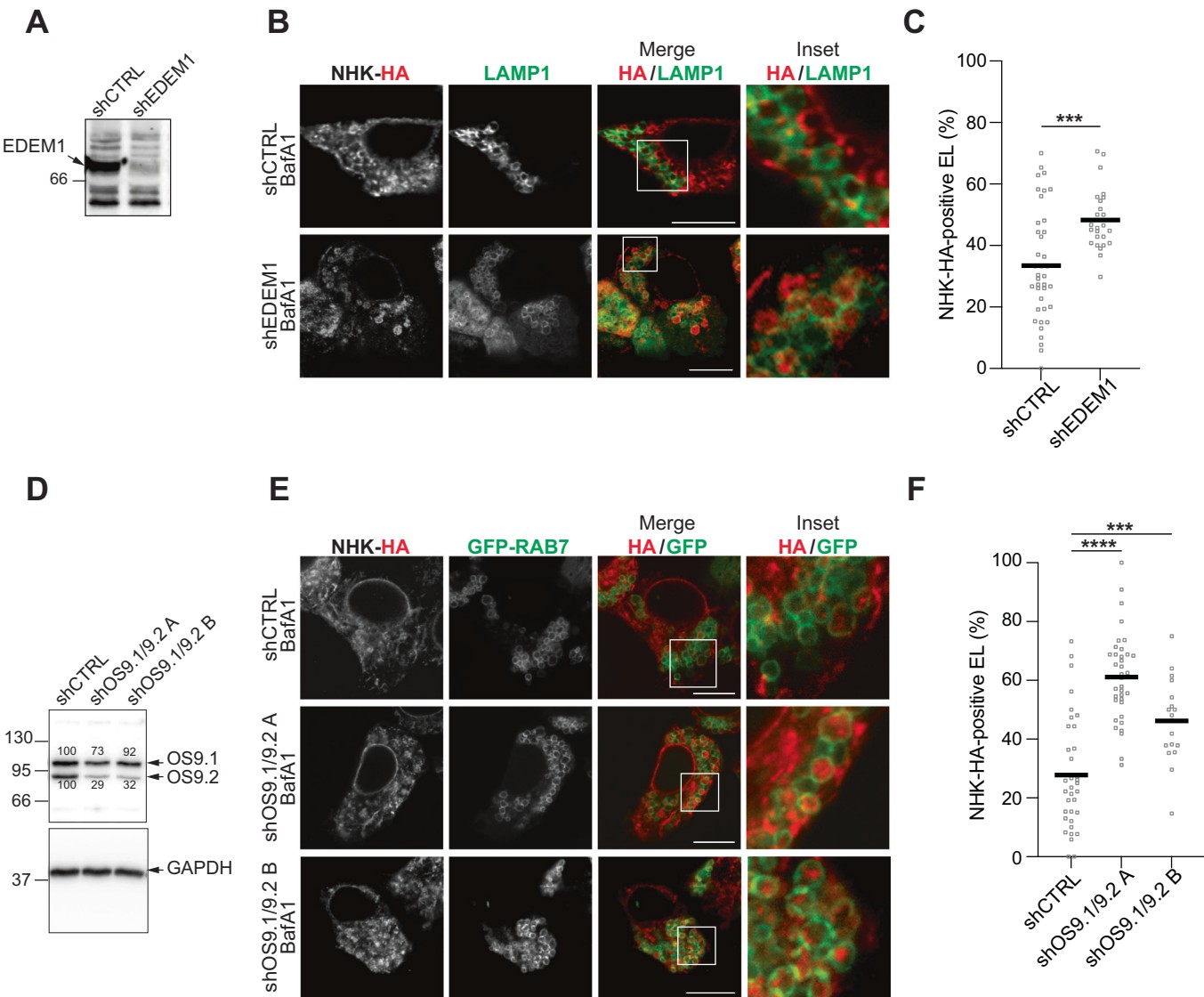

**Figure 3. The ERAD client NHK is re-directed to endolysosomes upon genetic inactivation of ERAD.**

(A) Western blot showing the efficiency of EDEM1 silencing in HEK293 cells. (B) Delivery of NHK in control cells (upper panels) and in cells with reduced ERAD upon EDEM1 silencing (lower panels). (C) LysoQuant quantification of (B) (n = 39 and 25 cells, respectively). (D) Western blot showing the efficiency of OS9.1/OS9.2 silencing. Two different oligos are used to reduce OS9.1/OS9.2 level. Quantification of OS9.1 and OS9.2 levels, normalized on GAPDH, is shown. (E) Delivery of NHK in control cells (upper panels) and in cells with reduced ERAD upon OS9.1/OS9.2 silencing (lower panels). (F) LysoQuant quantification of (E) (n = 35, 36, and 18 cells, respectively). Data Information: (C, F) Mean is shown. N = 3 independent experiments. (C) Unpaired t test, ***P < 0.001. (F) One-way analysis of variance (ANOVA) and Dunnett's multiple comparison test, ****P < 0.0001, ***P < 0.001. (B, E) Scale bars: 10 μM. Insets are shown with white squares. Source data are available online for this figure.

organelles is enhanced (Fig. 3E, panels in the middle, 3F) to higher levels compared to cells expressing shOS9.1/OS9.2 B (Fig. 3E, lower panels, F).

## FAM134B drives lysosomal delivery of NHK upon ERAD inhibition

Misfolded polypeptides that fail to engage the ERAD machinery (ATZ polymers are shown as examples in Fig. 2E,I) are segregated in ER subdomains displaying FAM134B at the limiting membrane and are eventually delivered to LAMP1-positive endolysosomal

compartments for clearance (Fregno et al, 2018, 2021; Rudinskiy and Molinari, 2023). If ERAD clients would co-opt the same pathway for lysosomal clearance upon ERAD inactivation, it is expected that NHK delivery to the LAMP1-positive compartment is substantially inhibited in cells lacking FAM134B. To test this hypothesis, the experiments described in the previous sections were reproduced in mouse embryonic fibroblasts (MEF) subjected to CRISPR/Cas9 genome editing to knockout FAM134B (Fig. 4A, lane 2). As shown above for other cell lines, also in wild-type (WT) MEF (Fig. 4B, upper panels, C) and in MEF lacking the FAM134B (Fig. 4B, lower panels, C) the ERAD client NHK is not delivered to

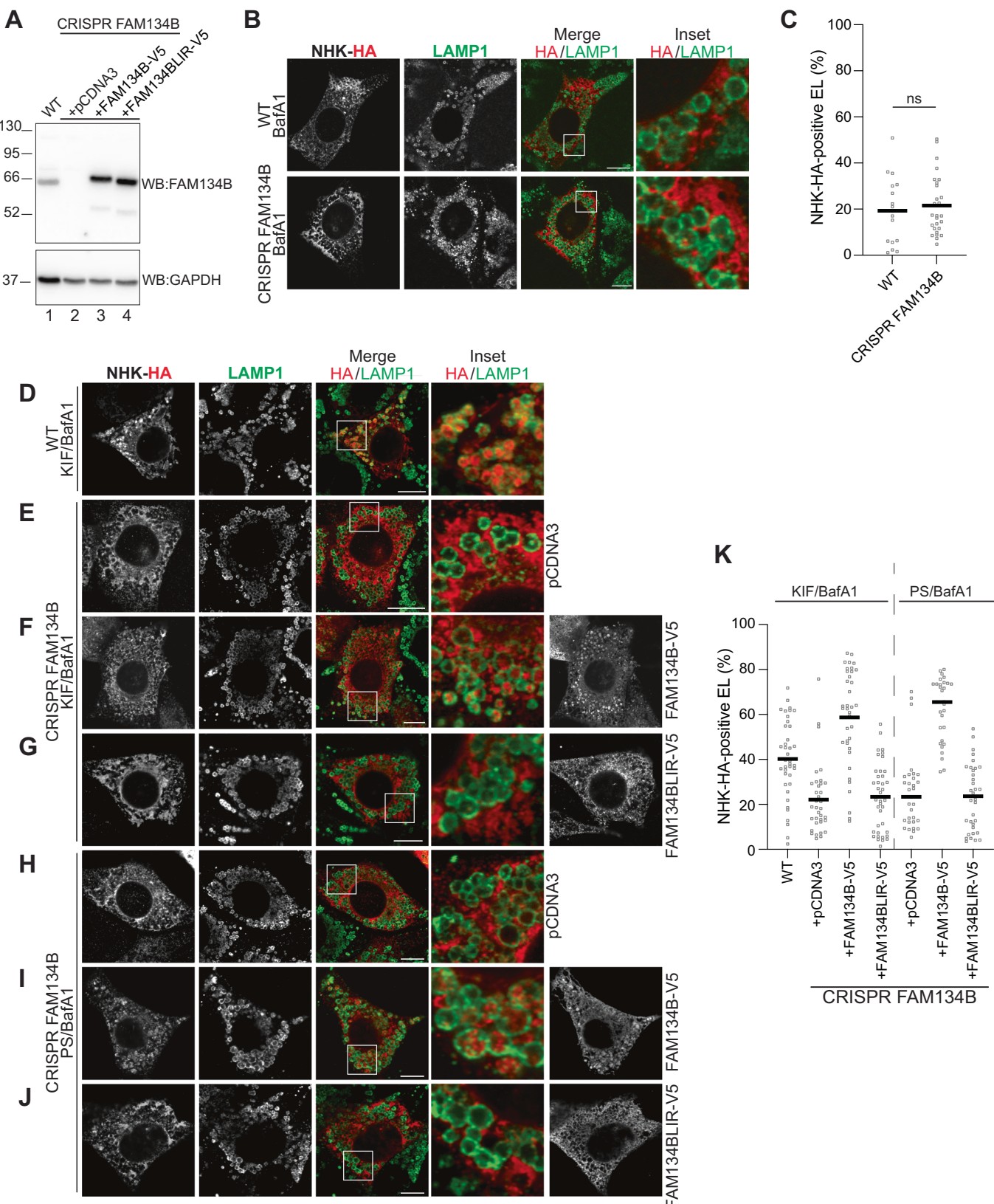

◄ **Figure 4.    FAM134B drives delivery of the ERAD client NHK into the ERLAD pathway.**

(A) Western blot showing deletion of endogenous FAM134B in MEF and levels of recombinant FAM134B-V5 and FAM134BLIR-V5 upon back-transfection. (B) NHK does not accumulate in LAMP1-positive endolysosomes in wild-type (upper panels) and FAM134B-KO cells (lower panels). (C) LysoQuant quantification of (B) (n = 17 and 27 cells, respectively). (D) NHK is delivered in LAMP1-positive endolysosomes upon ERAD inhibition with KIF in wild-type MEF. (E) NHK is not delivered in LAMP1-positive endolysosomes upon ERAD inhibition with KIF in MEF lacking FAM134B. (F) Back-transfection of FAM134B in FAM134B-KO MEF restores NHK delivery within endolysosomes. (G) Back-transfection of FAM134BLIR that cannot engage LC3 fails to restore NHK delivery within endolysosomes. (H–J) Analysis of NHK delivery to LAMP1-positive lysosomes in FAM134B-KO cells as in (E–G), when ERAD is inhibited with PS341. (K) LysoQuant quantification of (D–J) (n = 39, 35, 37, 39, 31, 30, and 35 cells, respectively). Data Information (C) Mean is shown. N = 3 independent experiments, unpaired t test, ⁿˢP > 0.05. (K) Mean is shown. N = 2 independent experiments. (B, D–J) Scale bars: 10 µM. Insets are shown with white squares. Source data are available online for this figure.

the LAMP1-positive endolysosomes. Inactivation of the ERAD pathway with KIF promotes NHK delivery to the LAMP1-positive degradative compartments in WT MEF (Fig. 4D,K), which is virtually abolished upon deletion of FAM134B (Fig. 4E,K, pCDNA3). The back-transfection of V5-tagged FAM134B in FAM134B-KO MEF (Fig. 4A, lane 3) restores NHK delivery to the LAMP1-positive endolysosomes (Fig. 4F,K). The back-transfection of FAM134BLIR (Fig. 4A, lane 4), an inactive variant of the ER-phagy receptor that carries mutations in the LIR domain preventing engagement of cytosolic LC3 molecules (Fregno et al, 2018), is not restoring NHK delivery to endolysosomes (Fig. 4G,K). The FAM134B-driven deviation of NHK into the ERLAD pathways is also induced when ERAD of NHK is abolished upon MEF exposure to the proteasomal inhibitor PS341 (Fig. 4H–K).

## Involvement of autophagy genes in compensatory ERLAD pathways

ER-phagy receptors control clearance of ER subdomains upon engagement of various autophagy gene products that will determine if ERLAD will proceed via *macro*-ER-phagy involving double-membrane autophagosomes (as for misfolded procollagen) (Forrester et al, 2019), or via other types of ER-phagy that will not involve autophagosomes and their biogenesis (as for ATZ) (Chino and Mizushima, 2023; Fregno et al, 2018; Fregno et al, 2021; Reggiori and Molinari, 2022). Previous work in our lab showed that FAM134B-controlled delivery of ATZ polymers to LAMP1-positive endolysosomes involves the LC3 lipidation machinery but does not involve the autophagosome biogenesis machinery. Consistently, deletion of the autophagy gene *Atg7* that abolishes LC3 lipidation (Komatsu et al, 2005) prevents ATZ delivery to endolysosomal compartments. In contrast, deletion of ATG13, a crucial component of the autophagosome biogenesis machinery (Hosokawa et al, 2009; Kaizuka and Mizushima, 2016; Suzuki et al, 2014) does not impact on ATZ delivery to the degradative district (Fregno et al, 2018). The repetition of these experiments to monitor the case of the ERAD client NHK confirms that the misfolded protein is not normally delivered to the LAMP1-positive endolysosomal compartment (Figs. 5A,H and Figs. 2–4), unless the ERAD pathway has been inactivated upon cell exposure to the α1,2-mannosidases inhibitor KIF or the proteasomal inhibitor PS341 (Figs. 5B,C,H and Figs. 2 and 4). In cells lacking ATG7, which show impaired generation of the lipidated form of LC3 (LC3-II, Fig. 5D), the delivery of NHK to the LAMP1-positive degradative compartment is abolished (Fig. 5E,H). In cells lacking ATG13 (Fig. 5F), the delivery of NHK to the LAMP1-positive degradative compartment proceeds unperturbed (Fig. 5G,H). These results recapitulate the phenotype previously observed for clearance of ATZ polymers,

which is hampered in cells with defective LC3 lipidation (Chu et al, 2014; Fregno et al, 2018; Hidvegi et al, 2010; Kroeger et al, 2009; Pastore et al, 2013; Sun et al, 2023; Teckman and Perlmutter, 2000), but remains unaffected in cells with defective autophagosome biogenesis (Fregno et al, 2018).

## Pharmacologic and genetic inhibition of ERAD triggers lysosomal delivery of BACE457Δ

Monitoring the fate of BACE457Δ, another folding-defective polypeptide that has been used extensively to characterized mechanistically the ERAD pathways (Bernasconi et al, 2010a; Bernasconi et al, 2010b; Cali et al, 2008; Eriksson et al, 2004; Horimoto et al, 2013; Molinari et al, 2003; Molinari et al, 2002; Ninagawa et al, 2011; Olivari et al, 2006; Olivari et al, 2005; Sokolowska et al, 2015), we confirmed that ERLAD acts as surrogate catabolic pathway to remove misfolded polypeptides from cells with dysfunctional ERAD (Fig. 6). As shown above for NHK, biochemical analyses of BACE457Δ decay confirm that the inhibition of lysosomal enzymes does not affect clearance from the ER (Molinari et al, 2002), unless cells have dysfunctional ERAD upon inhibition of cytosolic proteasomes with PS341 (Fig. 6A, lanes 6, 7 and gray zones in Fig. 6B) or of luminal mannosidases with KIF (Fig. 6A, lanes 10, 11 and gray zones in Fig. 6C). CLSM analyses confirm delivery of BACE457Δ within LAMP1-positive endolysosomes in cells with dysfunctional ERAD (Fig. 6D–G), and the involvement of FAM134B in the surrogate catabolic pathways activated under these circumstances (Fig. 6H–O).

## Discussion

A stringent protein quality control operates in the ER of eukaryotic cells. Proteins that have completed their folding program are released from ER-resident molecular chaperones and exit the ER to be delivered to their final intra- or extracellular destination. Folding is error-prone, and the rate of misfolding is substantially enhanced by mutations in the polypeptide sequence. The incapacity to fold correctly, results in selection of the aberrant gene product for degradation, or in the formation of aggregates that are retained in the biosynthetic compartment in soluble or insoluble forms. Dedicated machineries are available in the ER lumen and membranes to distinguish misfolded or incompletely folded polypeptides to be retained in the ER lumen, from native and functional proteins to be released (Ellgaard et al, 1999). Incompletely folded polypeptides retained in the ER are exposed to the folding environment and can eventually reach the native, transport-permissive architecture. When folding is impossible, the polypeptides are actively deviated into ERAD pathways that ensure their transport into the cytosol for

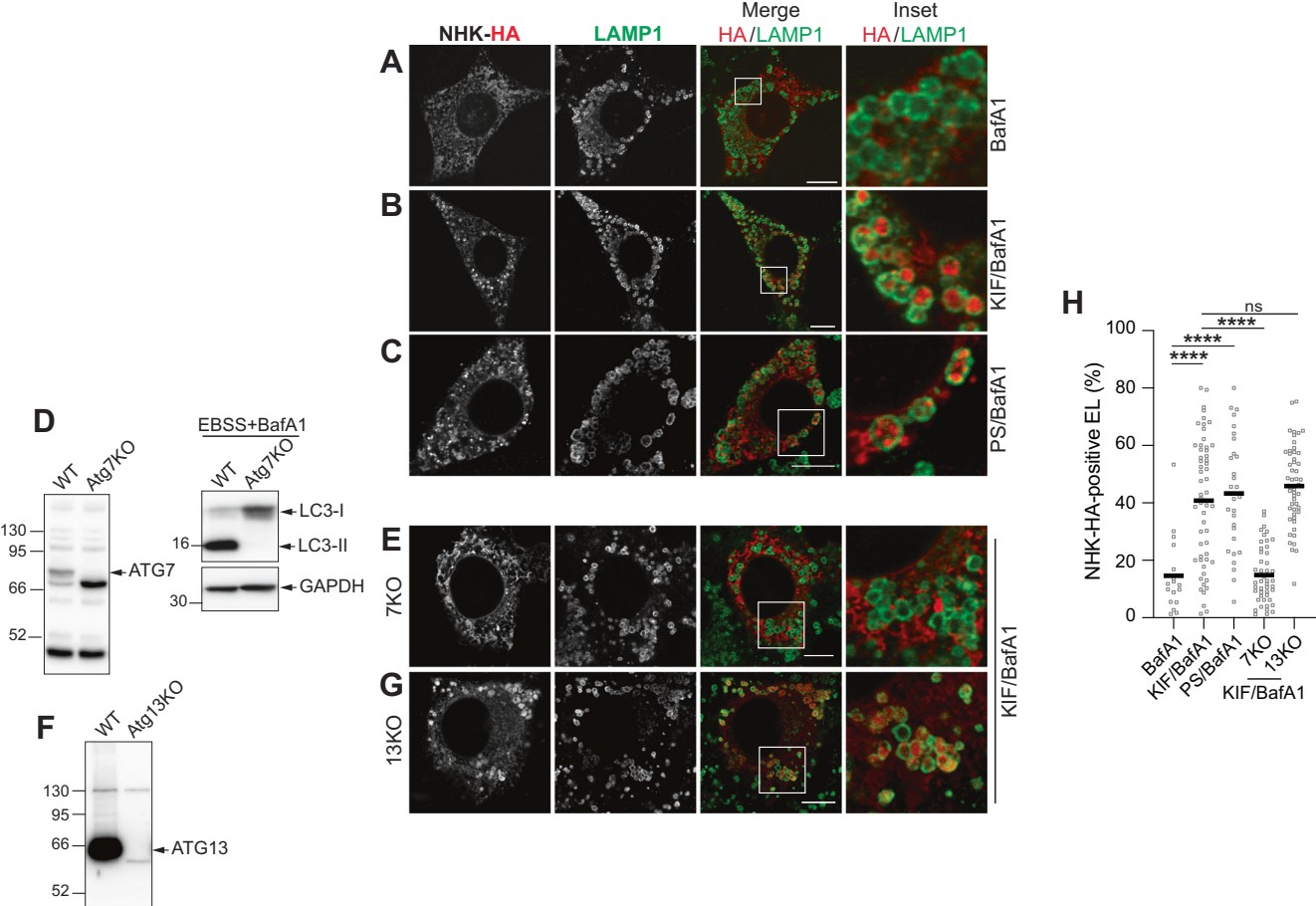

**Figure 5. The LC3 lipidation machinery is involved, and autophagosome biogenesis is dispensable, for delivery of NHK to LAMP1-positive endolysosomes upon ERAD inactivation.**

(A–C) Same as Fig. 2F–H, in MEF. (A) The ERAD client NHK is not delivered to LAMP1-positive endolysosomes. NHK is delivered and accumulates within endolysosomes inactivated with BafA1 upon ERAD inhibition with KIF (B) or with PS341 (C). (D) Western blots showing deletion of ATG7 (left panels) and defective LC3 lipidation in ATG7-KO upon starvation (right panels). (E) NHK does not accumulate in LAMP1-positive endolysosomes in ATG7-KO cells. (F) Western blot showing deletion of ATG13 in MEF. (G) NHK delivery within LAMP1-positive endolysosomes is not perturbed in ATG13-KO MEF. (H) LysoQuant quantification of (A–C, E, G) ($n$ = 17, 57, 29, 46, and 50 cells, respectively). Data Information: (H) Mean is shown. $N$ = 3 independent experiments. One-way analysis of variance (ANOVA) and Dunnett's multiple comparison test, $^{ns}P$ > 0.05, $^{****}P$ < 0.0001. (A–C, E, G) Scale bars: 10 μM. Insets are shown with white squares. Source data are available online for this figure.

proteasomal degradation (Fig. 1A). An increasing number of misfolded proteins is emerging in the literature that cannot enter ERAD pathways. In many cases, these are large polypeptides or polypeptides that are prone to form aggregates or polymers. A variety of ERLAD pathways are available in nucleated cells to segregate ERAD-resistant polypeptides in dedicated ER subdomains and to deliver them to endolysosomal/vacuolar degradative compartments (Fig. 1B) (Rudinskiy and Molinari, 2023).

For ERAD clients, numerous studies show that clearance from the ER is delayed, and not abolished upon inactivation of the ubiquitin-proteasome system. This also applies for cytosolic misfolded proteins, whose clearance from cells hosting inactive proteasomes has been proposed to rely on intervention of the giant protease tripeptidyl peptidase II, which shows enhanced activity in proteasome-inhibitor adapted cell (Geier et al, 1999; Glas et al, 1998; Tomkinson, 2019). Our study demonstrates that the intervention of ERLAD pathways is not limited to clearance of large proteins that fail to be dislocated across the ER membrane for

ERAD. Rather, ERLAD may be engaged by ERAD clients when their preferred road to destruction is dysfunctional or saturated by an excess of aberrant gene products (Fig. 7).

## Methods

### Expression plasmids and antibodies

HA-tagged ATZ and NHK and V5-tagged FAM134B and FAM134BLIR are subcloned in pcDNA3.1 plasmids. BACE457Δ is subcloned in pCDNA3 plasmid. pDest-EGFP-Rab7 is a kind gift from T. Johansen. The polyclonal anti-HA (Sigma cat. H6908), the anti-V5 (Invitrogen cat. 46-0705) and the anti-LAMP1 (Hybridoma Bank, 1D4B deposited by JT August) are described in (Fregno et al, 2021). The anti-FAM134B is a kind gift of M. Miyazaki (University of Colorado Denver); the anti-EDEM1 is from Sigma (cat. E8406); the anti-OS9 is from Novus Biologicals

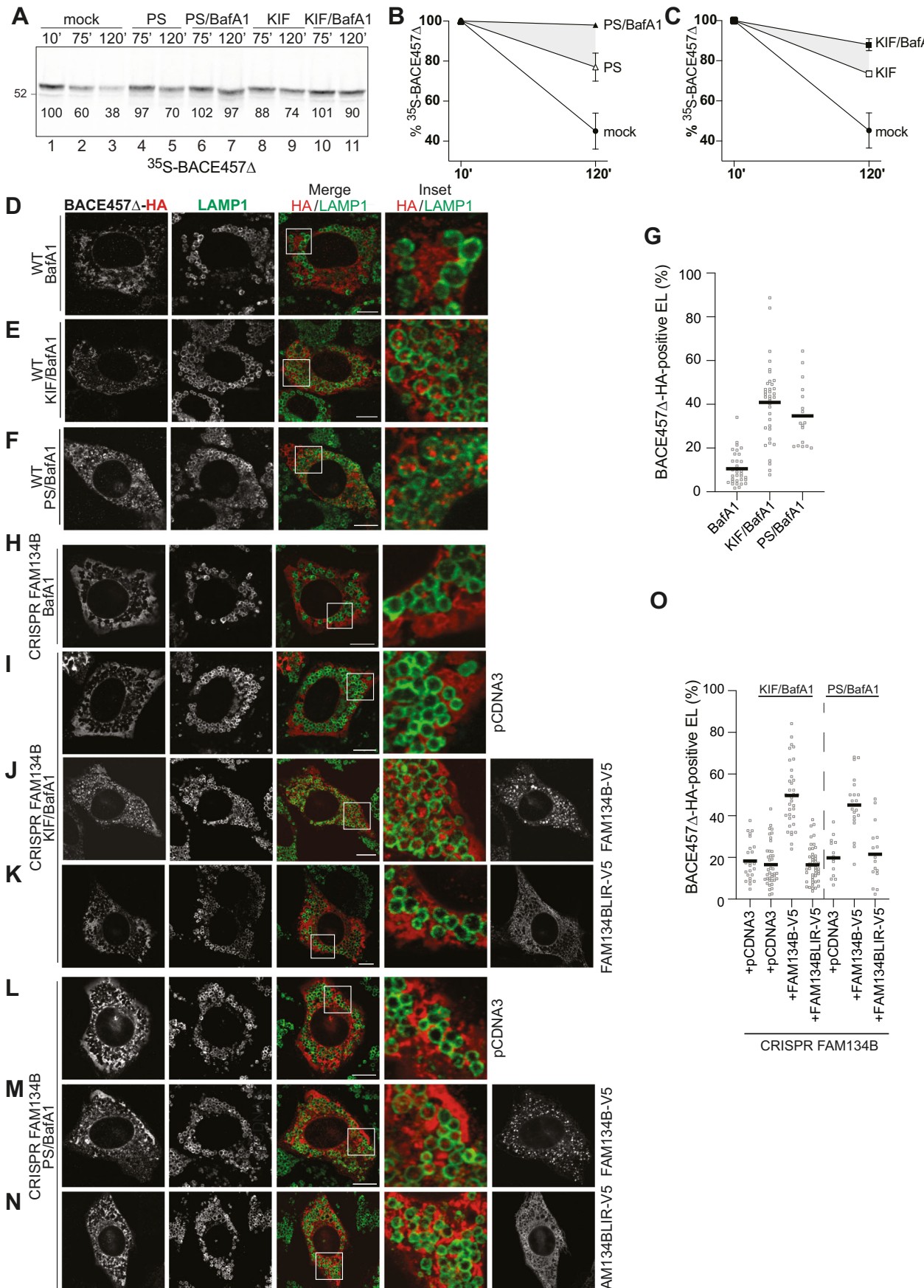

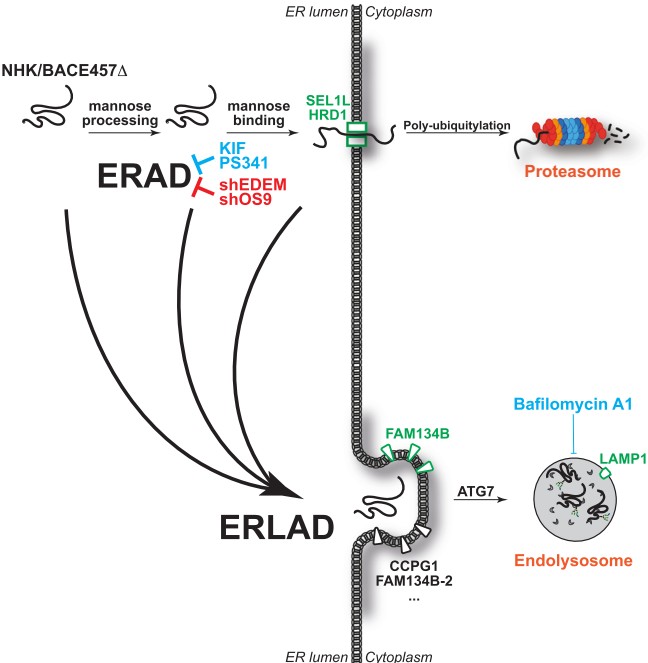

**Figure 7.** Schematic representation of how pharmacologic (blue) and genetic (red) ERAD perturbation triggers compensatory ERLAD programs.

Defective proteasomal clearance of NHK and BACE457Δ activates FAM134B-driven ERLAD. Other clients or maintenance of ER homeostasis in specific tissues may activate other ER-phagy receptors (e.g., CCPG1 in the pancreas and FAM134B-2 in the liver).

(cat. BC100-520); the anti-ATG7 and anti-ATG13 are from Sigma (cat. SAB4200304 and SAB4200100, respectively); the anti-LC3 from Novus (BC100-520); the anti-GAPDH from Merck (MAB374). Alexa-conjugated secondary antibodies are from Thermo Fisher; HRP-conjugated Protein A is from Invitrogen; HRP-conjugated anti-mouse is from SouthernBiotech).

## Cell Lines, transient transfections, pharmacologic inhibition

Flp-In™-3T3 cells (Thermo Fisher) stably expressing ATZ-HA or NHK-HA were generated following manufacturer instructions and cultured in DMEM supplemented with 10% FCS and 150 μg/ml

Hygromycin. MEF and HEK293 cells were grown in DMEM/10% FBS. HEK293FT cells expressing reduced levels of OS9.1 and OS9.2 are described in (Bernasconi et al, 2008). FAM134B-deficient MEF cells (CRISPR FAM134B) were generated using CRISPR/Cas9 genome editing protocol as described in (Fumagalli et al, 2016).

Transient transfections were performed using JetPrime transfection reagent (PolyPlus) following manufacturer's protocol. BafA1 (Calbiochem) was used at 50 nM for 12 h; KIF (Toronto Research Chemicals) was used at 200 μM for 12 h; PS341 (LubioScience) was used 12 h at 100 nM for Flp-In™-3T3 cells or at 5 nM for MEF cells. In wash/out experiments Flp-In™-3T3 cells were incubated for 8 h with 100 nM BafA1 and indicated drugs.

To induce autophagy in MEF WT and ATG7-KO, cells were washed three times with Earle's balanced salt solution (EBSS, Thermo Fisher) and then incubated for 4 h with 100 nM BafA1.

### Cell lysis, immunoprecipitation, and western blot

After treatments, cells were washed with ice-cold PBS containing 20 mM NEM and lysed with RIPA Buffer in HBS pH 7.4 supplemented with protease inhibitors. Post-nuclear supernatants (PNS) were collected after centrifugation at 10,600 × g for 10 min. For immunoprecipitations, PNSs were diluted with lysis buffer and incubated with Protein A (1:10 w/v, swollen in PBS) and select antibodies at 4 °C. After three washes of the immunoprecipitates with 0.5% Triton X-100 in HBS pH 7.4, beads were denatured for 5 min at 95 °C and subjected to SDS-PAGE. Proteins were transferred to PVDF membranes using the Trans-Blot Turbo Transfer System (Bio-Rad). Membranes were blocked with 10% (w/v) non-fat dry milk (Bio-Rad) in TBS-T and stained with primary antibodies diluted in TBS-T followed by HRP-conjugated secondary antibodies or Protein A diluted in TBS-T. Membranes were developed using Western Bright ECL or Quantum (Advansta), and signals captured on Fusion FX (Vilber). Images were quantified with the Evolution Capture Edge (Vilber).

### Metabolic labeling

Seventeen hours after transient transfections, cells were pulsed with 0.05 mCi [35 S]methionine/cysteine mix and chased for the indicated time points with DMEM supplemented with 5 mM cold methionine and cysteine. Cells were detergent-solubilized and radiolabeled proteins were revealed with Typhoon FLA 9500, version 1.0 (GE Healthcare). Radioactive signals were quantified using the ImageQuant software (Molecular Dynamics, GE Healthcare).

## Confocal laser scanning microscopy

Cells plated on Alcian Blue-treated glass coverslips were washed with PBS and fixed at room temperature for 20 min in 3.7% formaldehyde diluted in PBS. Cells were permeabilized for 15 min with 0.05% saponin, 10% goat serum, 10 mM HEPES, 15 mM glycine (PS) and exposed for 90 min to primary antibodies diluted 1:100 in PS. After washes with PS, cells were incubated for 45 min with Alexa Fluor-conjugated secondary antibodies diluted 1:300 in PS. Cells were rinsed with PS and water and mounted with Vectashield (Vector Laboratories). Confocal images were acquired on a Leica TCS SP5 microscope with a Leica HCX PL APO lambda blue 63.0 × 1.40 OIL UV objective. IF images were acquired by two different people, double-blinded, and raw data were quantified using LysoQuant (Morone et al, 2020).

## Statistical analyses

Plots and statistical analyses were performed using GraphPad Prism10 (GraphPad Software Inc.). In this study, one-way ANOVA with Dunnett's multiple comparisons test and unpaired $t$ test were used to assess statistical significance. An adjusted $P$ value < 0.05 was considered as statistically significant.

## Data availability

This study includes no data deposited in external repositories.

The source data of this paper are collected in the following database record: biostudies:S-SCDT-10_1038-S44319-024-00165-y.

## Peer review information

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

## Acknowledgements

The authors thank members of Molinari's group for discussions and the IRB Microscopy core facility for assistance with microscopy. MM is supported by Swiss National Science Foundation grants 310030_214903 and 320030_227541, Eurostar E!2228, Innosuisse 35449.1 IP-LS.

## Author contributions

**Elisa Fasana**: Data curation; Formal analysis; Investigation; Methodology; Writing—review and editing. **Ilaria Fregno**: Data curation; Formal analysis; Investigation; Methodology; Writing—review and editing. **Carmela Galli**: Data curation; Formal analysis; Investigation; Methodology; Writing—review and editing. **Tatiana Soldà**: Formal analysis; Investigation; Methodology; Writing—review and editing. **Maurizio Molinari**: Data curation; Formal analysis; Supervision; Funding acquisition; Investigation; Methodology; Writing—original draft; Project administration; Writing—review and editing.

Source data underlying figure panels in this paper may have individual authorship assigned. Where available, figure panel/source data authorship is listed in the following database record: biostudies:S-SCDT-10_1038-S44319-024-00165-y.

## Disclosure and competing interests statement

The authors declare no competing interests.

