## [Peer Review File · EMBO Reports]

ER-to-lysosome-associated degradation acts as failsafe mechanism upon ERAD dysfunction

Elisa Fasana, Ilaria Fregno, Carmela Galli, Tatiana Soldà and Maurizio Molinari

Corresponding author(s): Maurizio Molinari (maurizio.molinari@irb.usi.ch)

Review Timeline:

Transfer Date:	1st Mar 24
Editorial Decision:	27th Mar 24
Revision Received:	8th Apr 24
Editorial Decision:	23rd Apr 24
Revision Received:	25th Apr 24
Accepted:	30th Apr 24

Transaction Report: This manuscript was transferred to EMBO reports following peer review at Review Commons.

Review
COMMONS

Review #1

1. Evidence, reproducibility and clarity:

Evidence, reproducibility and clarity (Required)

In this manuscript from Fasana et al., the authors present data that investigates potential compensatory degradation pathways for misfolded glycoproteins in the ER - postulating that the ER-to-lysosome associated degradation (ERLAD) pathway becomes employed in the absence of a path for substrates to reach the ER-associated degradation (ERAD) mechanism. Using the classic ERAD substrate alpha1-antitrypsin NHK variant (NHK), the authors first demonstrate that pharmacologically preventing access of NHK to ERAD either with KIF (early) or PS-341 (late) elevates the number of LAMP-1 positive endolysosomes also immunoreactive for NHK (via HA), similar to what is observed for the ATZ variant that forms polymers in the ER (Fig 2). The authors next use shRNAs that silence essential ERAD factors (EDEM1, OS-9) involved in glycan recognition to demonstrate comparable enrichment of NHK in endolysosomes through genetic disruption (Fig 3). Next, the authors employ FAM134B-deficient MEFs to demonstrate the requirement for this ER-phagy receptor when ERAD is unavailable (Fig 4). Reconstituting FAM134B^{-/-} MEFs treated with KIF/PS-341 + Baf, with a full length FAM134B rescue plasmid restores endolysosomal accumulation of NHK while a FAM134B-ΔLIR does not, providing supporting evidence for substrate rerouting to ERLAD. Finally, the authors use knockouts of Atg7 and Atg13 to demonstrate dependence on LC3 lipidation and independence from macro-ERphagy (Fig 6), that points towards a pathway that is like that used to remove ATZ polymers. From these data, the authors conclude that ERLAD is increasingly engaged for substrate degradation when ERAD is impaired.

****Major comments:****

1. All assays rely on quantification of the NHK-HA substrates by microscopy. Would it be possible for the authors to also include biochemical analysis of NHK - potentially including data assessing its changing abundance and glycosylation state?
2. In Figure 3D, the knockdown of OS-9.1/2 is modest compared to that of EDEM1 (Fig 3A). Moreover, there is only data from single shRNAs presented. Could the authors please at least include another shRNA to confirm and demonstrate whether the targeting to ERLAD is accordingly scaled to loss of access to ERAD (based on the degree of OS-9 or EDEM1 remaining)?
3. While degradation is implied, it is not specifically demonstrated at any point in the manuscript. Perhaps the authors might include some demonstration of NHK

stabilization in one of the figures via a translational shutoff or pulse-chase assay.

4. 10-30% of NHK-HA positive endolysosomes are detected even with Baf alone (e.g. Fig 2E)? Does this mean that Baf impairs ERAD to some extent since or is it evidence for continuous ERLAD involvement when ERAD is intact? If so, how much is its contribution?

5. An accounting of how much ERLAD is contributing to NHK degradation with or without ERAD impairment is not really present.. Effectively, how much degradation capacity is ERLAD making up? These would be interesting data to include if possible as they would speak to the "division of labour" for ER substrate degradation its potentially dynamic nature.

****Minor comments:****

1. In Figure 4, an increase is observed for the rescue of FAM134B^{-/-}-MEFs with WT FAM134B that is 50% greater than that of WT MEFs, suggesting that its availability might be rate limiting. Could the authors compare the relative levels of FAM134B for the WT and KO-rescue MEFs to address this possibility?

2. In Figures 1 and 6, the terms siOS9 and siEDEM1 are used but Figure 3 shows data from shRNAs and not siRNAs.

3. Samples from Figure 3 treated with Baf but this is not indicated in the figure or figure legend.

4. VCP/p97 inhibitors typically stabilize ERAD glycoprotein substrates better than proteasome inhibitors do. Is the same degree of endolysosomal targeting present

2. Significance:

Significance (Required)

Deconvolution of the different pathways taken by misfolded proteins to escape the ER is of great interest not only to the ER community but also represents consequences to consider for those interested in therapeutics involving UPS inhibition. While concise, this manuscript does a good job of trying to demonstrate the principal of substrate rerouting and the prioritisation of degradation pathways. Overall, the manuscript is well written, the experiments presented are performed to a sufficient standard, the data are lean but of good quality, and the appropriate statistical analyses have mostly been included where necessary and are described. The Methods and Materials is brief but describes the experiments that have been performed. The manuscript is brief in its results and would obviously benefit from additional complementary assays that would strengthen and broaden the authors arguments for rerouting. But too their credit, the authors do not grossly overstate their findings and merely present the culmination of a set of experiments to answer a single question - what happens to a misfolded

glycoprotein substrate when ERAD is impaired. This is a key question with broad implications.

While their limited data clearly demonstrates an acquired dependence on ERLAD, one can't help but wonder how broadly these findings hold true, as only a single glycoprotein substrate example is used. Moreover, it is not clear what percentage ERLAD contributes to overall NHK degradation (with or without ERAD) as the total NHK amount remaining is not assessed or measured. Nevertheless, the manuscript is an advancement of understanding of the fate of substrates unable to access ERAD and raises many future questions of interdependency between the ERAD and ERLAD pathways. The data just need a bit of shoring up.

Expertise - ERAD, UPS, protein quality control

3. How much time do you estimate the authors will need to complete the suggested revisions:

Estimated time to Complete Revisions (Required)

(Decision Recommendation)

Between 3 and 6 months

Yes

Review #2

1. Evidence, reproducibility and clarity:

Evidence, reproducibility and clarity (Required)

The endoplasmic reticulum (ER) is a crucial site for protein synthesis and folding within the cell, and strict protein quality control is essential for maintaining ER homeostasis. In this context, ER-associated degradation (ERAD) and the unfolded protein response (UPR) play pivotal roles. Recent researches have highlighted the significance of ER-phagy in protein quality control. In this manuscript, the authors demonstrate the role of FAM134B in degrading misfolded proteins such as ATZ through the ER-phagy pathway when the ERAD pathway is obstructed. This work partially addresses a prominent issue in the field, unveiling the interconnections between different regulatory pathways in maintaining ER homeostasis.

Major issues:

1. In a multitude of experiments, the authors employed Bafilomycin A1 (BafA1) to block the fusion between autophagosomes and lysosomes, attempting to demonstrate that the clearance of misfolded proteins mediated by FAM134B is independent of autolysosomes. However, in Figure 4, the lack of rescue of FAM134B knockout by overexpressing FAM134B Δ LIR suggests a dependence on the interaction between FAM134B and LC3. The conclusions drawn before and after appear contradictory.
2. Some Western blot data are insufficient to substantiate the author's conclusions. For instance, in Figure 5D, the ATG7 KO line is inadequately supported
3. The author employed Lamp1 antibody for lysosomal staining in cells and observed a significant abundance of lysosomes in some experiments, as depicted in Figure 2C, 2D, 4I, etc. Is the phenomenon of lysosomes extensively filling the entire cell a common occurrence? Is it indicative of a normal physiological state?

Minor issues:

1. Some immunofluorescence experimental data are unclear. Please request the authors to replace these with more distinct images, as seen in Figure 3B and 3E.
2. Some expressions appear to be questionable. For instance, the necessity of utilizing endolysosomes requires clarification.
3. Some writing lacks precision, such as referring to FAM134B as FAM134.

2. Significance:

Significance (Required)

- *General assessment:*
- *Advance:* provide an meaningful evidence that how two degradative pathways are coordinated in maintaining ER homeostasis.
- *Audience:* cell biologist
- *Reviewer's expertise:* autophagy, vesicle trafficking, organelle biology

3. How much time do you estimate the authors will need to complete the suggested revisions:

Estimated time to Complete Revisions (Required)

(Decision Recommendation)

Between 1 and 3 months

No

Review #3

1. Evidence, reproducibility and clarity:

Evidence, reproducibility and clarity (Required)

In their study, Fasana and colleagues investigate protein quality control in the ER. Specifically, they test whether folding-incompetent proteins that are normally cleared

by ER-associated degradation (ERAD) can also be targeted for degradation by direct vesicular transport from the ER to lysosomes in case ERAD is blocked. They show that blocking ERAD pharmacologically or genetically indeed leads to re-rerouting of an ERAD model substrate (the NHK variant of alpha-antitrypsin) to lysosomes and that this pathway requires the reticulon-like protein FAM134B, the ability of FAM134B to interact with the ubiquitin-like protein LC3 and the machinery for LC3 lipidation.

The paper is, for the most part, easy to follow. There are, however, a few minor issues and I think the authors could do more to connect their work with similar studies in the literature. Accordingly, I have some general and specific suggestions to make the manuscript more accessible for the reader.

****General suggestions****

1. To avoid confusion, it would be helpful to more clearly distinguish between vesicular transport to endolysosomes and autophagy. Previous work by the authors has defined a trafficking pathway from the ER to endolysosomes that appears to rely on conventional vesicle-mediated transport (Fregno et al, EMBO J 2018). This pathway delivers material from the ER lumen to the lumen of endolysosomes, which are both topologically equivalent to the extracellular space. Hence, this pathway is distinct from autophagy, which is the transport of cytoplasmic components to endolysosomes and thus the transport of material from intracellular to extracellular space. This distinction is particularly important as both vesicular ER-to-lysosome transport and autophagy of the ER involve LC3 and FAM134B, which is typically referred to as an ER-phagy receptor. To make this less confusing, it may be helpful to explain that FAM134B appears to be a multifunctional molecule that can function as a receptor for macroautophagy but also in the vesicular transport pathway studied here. In addition, it would be helpful to point out that LC3 appears to also have roles unrelated to autophagosome formation.
2. Several recent papers that appear relevant to the present study are not mentioned. In particular, Sun et al., Dev Cell 2023 (PMID: 37922908) appears worthy of discussion, as does Gonzalez et al., Nature 2023 (PMID: 37225996).

****Specific suggestions****

1. Abstract: The abstract begins with "About 40% of the eukaryotic cell's proteome is synthesized ... in the ER." Similar statements can be found in many papers and purportedly reflect common knowledge. However, it is unclear where the figure of 'about 40%' comes from. It would be proper to provide a reference and demonstrate

that giving such a fairly precise estimate is supported by experimental data.

Alternatively, the statement could be modified to avoid being precise than is justified.

2. p2: "The ER is site of gene expression in nucleated cells and ... native proteins to be delivered at their site of activity ...". There is something missing at the beginning of this sentence. Also, it should be 'delivered to their site of activity', not 'delivered at'.

3. p2: "... by mechanistically distinct ER-phagy pathways collectively defined as ER-to-lysosome-associated degradation ERLAD." This statement suggests that all pathways subsumed under the term ERLAD are ER-phagy pathways, which I believe is misleading (see comment above on the distinction between autophagy and vesicular transport pathway).

4. p2: "KIF selectively ...". Please spell out KIF and explain what kind of compound it is.

5. p3: "Notably, ERAD inhibition delays, rather than blocking degradation of ERAD clients ...". Please correct, for example: Notably, ERAD inhibition delays rather than blocks degradation of ERAD clients ...

6. Figures 2 - 5: The number of quantified cells is given but it is not clear if experiments were done once or in biological replicates. Please indicate this in the figure legends.

7. p4: "To verify if ERAD inactivation ..." sounds odd. Less ambiguous would be 'To test whether' or 'To ask if'.

8. p7, beginning of discussion: Please correct "delivered at" to 'delivered to'.

2. Significance:

Significance (Required)

This is a concise and convincing manuscript with a clear message. The idea that proteins that cannot be processed by ERAD can be eliminated by other means, for instance by autophagy, is not new. Similarly, the FAM134B- and LC3-dependent pathway for ER-to-lysosome transport has been described by the authors before (Fregno et al, EMBO J 2018). Furthermore, the study exclusively relies on microscopy and does not attempt to tackle new mechanistic questions. Still, this study presents a definite functional advance in our understanding of the interplay of various ER quality control pathways.

The findings presented here will be of interest mainly to molecular cell biologists working on protein quality control and organelle homeostasis. However, given the disease-relevance of misfolded proteins, and alpha-antitrypsin in particular, the impact of this study may eventually go beyond basic research and may also interest translational researchers.

3. How much time do you estimate the authors will need to complete the suggested revisions:

Estimated time to Complete Revisions (Required)

(Decision Recommendation)

Less than 1 month

Yes

Full Revision

Manuscript number: #RC-2023-02281

Corresponding author(s): Maurizio Molinari

Point-by-point description of the revisions

This section is mandatory. Please insert a point-by-point reply describing the revisions that were already carried out and included in the transferred manuscript.

Reviewer #1 (Evidence, reproducibility and clarity (Required)):

In this manuscript from Fasana et al., the authors present data that investigates potential compensatory degradation pathways for misfolded glycoproteins in the ER - postulating that the ER-to-lysosome associated degradation (ERLAD) pathway becomes employed in the absence of a path for substrates to reach the ER-associated degradation (ERAD) mechanism. Using the classic ERAD substrate alpha1-antitrypsin NHK variant (NHK), the authors first demonstrate that pharmacologically preventing access of NHK to ERAD either with KIF (early) or PS-341 (late) elevates the number of LAMP-1 positive endolysosomes also immunoreactive for NHK (via HA), similar to what is observed for the ATZ variant that forms polymers in the ER (Fig 2). The authors next use shRNAs that silence essential ERAD factors (EDEM1, OS-9) involved in glycan recognition to demonstrate comparable enrichment of NHK in endolysosomes through genetic disruption (Fig 3). Next, the authors employ FAM134B-deficient MEFs to demonstrate the requirement for this ER-phagy receptor when ERAD is unavailable (Fig 4). Reconstituting FAM134B^{-/-} MEFs treated with KIF/PS-341 + Baf, with a full length FAM134B rescue plasmid restores endolysosomal accumulation of NHK while a FAM134B-ΔLIR does not, providing supporting evidence for substrate rerouting to ERLAD. Finally, the authors use knockouts of Atg7 and Atg13 to demonstrate dependence on LC3 lipidation and independence from macro-ERphagy (Fig 6), that points towards a pathway that is like that used to remove ATZ polymers. From these data, the authors conclude that ERLAD is increasingly engaged for substrate degradation when ERAD is impaired.

MAJOR COMMENTS

1. All assays rely on quantification of the NHK-HA substrates by microscopy. Would it be possible for the authors to also include biochemical analysis of NHK - potentially including data assessing its changing abundance and glycosylation state?

To consider this, and other comments, the new submission includes biochemical data (pulse-chase analyses) on NHK (new panels A-D in Fig. 2) and on BACE457delta, an additional ERAD substrate (new Fig. 6). Please also refer to Comment 3.

2. In Figure 3D, the knockdown of OS-9.1/2 is modest compared to that of EDEM1 (Fig 3A). Moreover, there is only data from single shRNAs presented. Could the authors please at least include another shRNA to confirm

and demonstrate whether the targeting to ERLAD is accordingly scaled to loss of access to ERAD (based on the degree of OS-9 or EDEM1 remaining)?

The reviewer is right. The phenotype (i.e., lysosomal delivery of NHK, Figs. 3B, 3C) is quite modest upon EDEM1 silencing. However, one has to consider that in contrast to OS9 lectins, EDEM1 is an enzyme, and residual protein may partially facilitate NHK de-mannosylation and access to the ERAD pathways and therefore reduce the ERLAD contribution for NHK clearance in these cells. Moreover, cells also express EDEM2 and 3 that may partially compensate the loss of EDEM1.

3. While degradation is implied, it is not specifically demonstrated at any point in the manuscript. Perhaps the authors might include some demonstration of NHK stabilization in one of the figures via a translational shutoff or pulse-chase assay.

In the new submission, we show biochemical analyses (pulse-chase) that reveal the decay of radiolabeled NHK (Fig. 2A, lanes 1-3) and BACE457delta (Fig. 6A, lanes 1-3), the inhibition by PS341 (lanes 4, 5) and by KIF (lanes 8, 9), and the intervention of lysosomal enzymes when ERAD is inhibited (lanes 6, 7 and 10, 11). Moreover, we confirm that the protein delivered to the endolysosome is eventually degraded by performing a Bafilomycin washout experiment (new Fig. 2J-2O).

4. 10-30% of NHK-HA positive endolysosomes are detected even with Baf alone (e.g. Fig 2E)? Does this mean that Baf impairs ERAD to some extent since or is it evidence for continuous ERLAD involvement when ERAD is intact? If so, how much is its contribution?

Pulse-chase analyses (new Fig. 2D) and published data show that BafA1 or chloroquine do not inhibit clearance of the ERAD substrates NHK and BACE457delta (e.g., Liu et al 1999, Molinari et al 2002, references in the manuscript). A basal level of endolysosomal delivery between the 20 and 30% as quantified with LysoQuant is observed in all experiments (Figs. 2I, 2O, 3C, 3F, 4C, 4K, 5H, 6G, 6O), which have been performed in 3 different cell lines (3T3, HEK293, MEF). We measure similar basal levels also when ER-phagy is monitored on quantification of lysosomal delivery of endogenous ER marker proteins (e.g., CNX), possibly to be ascribed to constitutive ER phagy that controls physiologic ER turnover.

5. An accounting of how much ERLAD is contributing to NHK degradation with or without ERAD impairment is not really present.. Effectively, how much degradation capacity is ERLAD making up? These would be interesting data to include if possible as they would speak to the "division of labour" for ER substrate degradation its potentially dynamic nature.

The biochemical analyses show the contribution of ERLAD on NHK (new Figs 2B, 2C, grey zones) and BACE457delta (new Figs. 6B,C, grey zones) clearance, when ERAD is dysfunctional.

MINOR COMMENTS

1. In Figure 4, an increase is observed for the rescue of FAM134B^{-/-}-MEFs with WT FAM134B that is 50% greater

that of WT MEFs, suggesting that its availability might be rate limiting. Could the authors compare the relative levels of FAM134B for the WT and KO-rescue MEFs to address this possibility?

The referee is right in assuming that FAM134B, expressed at low levels in these cells, is limiting. We now show the levels of endogenous FAM134B and of recombinant FAM134B in WB (new Fig. 4A).

2. In Figures 1 and 6, the terms siOS9 and siEDEM1 are used but Figure 3 shows data from shRNAs and not siRNAs.

We apologize for the mistake. We have corrected this in the new Figures 1 and 7.

3. Samples from Figure 3 treated with Baf but this is not indicated in the figure or figure legend.

We have corrected this, thank you.

4. VCP/p97 inhibitors typically stabilize ERAD glycoprotein substrates better than proteasome inhibitors do. Is the same degree of endolysosomal targeting present ?

For the convenience of the reviewer (we did not put these data in the new manuscript). In our experiments, the p97 inhibitor DBE-Q is less efficient in deviating NHK to the endolysosomal degradative compartments, if compared with KIF (see below). At higher doses, DBE-Q also inhibits other AAA-ATPases (e.g., VPS4, which plays a role in certain types of autophagy). This, or other cross-reactivities of DBE-Q could explain the moderate capacity to activate ERLAD pathways as a response of ERAD inhibition, if compared with the phenotypes observed when ERAD is inhibited with KIF or PS341.

Reviewer #1 (Significance (Required)):

Deconvolution of the different pathways taken by misfolded proteins to escape the ER is of great interest not only to the ER community but also represents consequences to consider for those interested in therapeutics involving UPS inhibition. While concise, this manuscript does a good job of trying to demonstrate the principal of substrate rerouting and the prioritisation of degradation pathways. Overall, the manuscript is well written, the experiments presented are performed to a sufficient standard, the data are lean but of good quality, and the appropriate statistical analyses have mostly been included where necessary and are described. The Methods and Materials is brief but describes the experiments that have been performed. The manuscript is brief in its results and would obviously benefit from additional complementary assays that would strengthen and broaden the authors arguments for rerouting. But too their credit, the authors do not grossly overstate their findings and merely present the culmination of a set of experiments to answer a single question - what happens to a misfolded glycoprotein substrate when ERAD is impaired. This is a key question with broad implications.

While their limited data clearly demonstrates an acquired dependence on ERLAD, one can't help but wonder how broadly these findings hold true, as only a single glycoprotein substrate example is used.

We have now added a complete set of experiments (imaging + biochemical to monitor clearance of the model polypeptides by pulse-chase analyses) performed with a second ERAD substrate (BACE457delta, Fig. 6). These data fully recapitulate the results obtained with NHK.

Moreover, it is not clear what percentage ERLAD contributes to overall NHK degradation (with or without ERAD) as the total NHK amount remaining is not assessed or measured.

Pulse-chase analyses (new Fig. 2D) and published data (e.g., Liu et al 1999, Molinari et al 2002, references in the manuscript) show that BafA1 or chloroquine do not inhibit clearance of the ERAD substrates NHK and BACE457delta. The biochemical analyses now show the contribution of ERLAD on NHK (new Figs 2B, 2C, grey zones) and BACE457delta (new Figs. 6B,C, grey zones) clearance, when ERAD is dysfunctional.

Nevertheless, the manuscript is an advancement of understanding of the fate of substrates unable to access ERAD and raises many future questions of interdependency between the ERAD and ERLAD pathways. The data just need a bit of shoring up.

Expertise - ERAD, UPS, protein quality control

Reviewer #2 (Evidence, reproducibility and clarity (Required)):

The endoplasmic reticulum (ER) is a crucial site for protein synthesis and folding within the cell, and strict protein quality control is essential for maintaining ER homeostasis. In this context, ER-associated degradation (ERAD) and the unfolded protein response (UPR) play pivotal roles. Recent researches have highlighted the significance of ER-phagy in protein quality control. In this manuscript, the authors demonstrate the role of FAM134B in degrading misfolded proteins such as ATZ through the ER-phagy pathway when the ERAD pathway is obstructed. This work partially addresses a prominent issue in the field, unveiling the interconnections between different regulatory pathways in maintaining ER homeostasis.

Major issues:

1: In a multitude of experiments, the authors employed Bafilomycin A1 (BafA1) to block the fusion between autophagosomes and lysosomes, attempting to demonstrate that the clearance of misfolded proteins mediated by FAM134B is independent of autolysosomes. However, in Figure 4, the lack of rescue of FAM134B knockout by overexpressing FAM134B Δ LIR suggests a dependence on the interaction between FAM134B and LC3. The conclusions drawn before and after appear contradictory.

We apologize if our explanations were unclear. We have now modified the text and performed new experiments to clarify these issues.

The inhibitor of the V-ATPase BafA1 is used here to inhibit the activity of lysosomal hydrolases and to accumulate undegraded material in the LAMP1-endolysosomes (note that these endolysosomes also display RAB7 at their limiting membrane) (Fregno et al 2018, Forrester et al 2019, Fregno et al 2021, ...).

In Figs. 2A-2D, we now monitor the lack of NHK stabilization by cell exposure to BafA1 (Fig. 2D), which correlates with lack of accumulation of NHK in the LAMP1-positive compartment (e.g., Fig. 2F, 2J, and quantifications in 2I and 2O). The biochemical data also show that BafA1 stabilizes NHK in cells where ERAD has been inactivated with PS341 or KIF (Fig. 2A, lanes 6, 7, 10, 11 and grey zones in Figs. 2B and 2C), which correlates with accumulation of NHK in LAMP1-positive organelles (Figs. 2G, 2H, 2I, 2K, 2M, 2O).

In Figs. 2J-2O, we have now added panels showing that NHK clearance from the LAMP1-positive endolysosome lumen is restored upon BafA1 washout.

Importantly, the involvement of the lipidation machinery, of the ER-phagy receptor FAM134B and of the LC3-binding function of FAM134B (the LIR), does not necessarily imply the involvement of autophagosomes in the process under investigation, as the comment by the referee seems to suggest. For example, both the clearance from the ER of ATZ polymers and of mutant forms of procollagen rely on the LC3 lipidation machinery and on the LC3-binding function of FAM134B, but ERLAD of ATZ polymers does not rely on autophagosomes intervention (new Fig. 1B, arrow 1 and Fregno et al 2018), whereas ERLAD of procollagen relies on intervention of autophagosomes (new Fig. 1B, arrow 2 and Forrester et al 2019).

2: Some Western blot data are insufficient to substantiate the author's conclusions. For instance, in Figure 5D, the ATG7 KO line is inadequately supported

The WB shows the absence of ATG7 in the ATG7-KO cells (a well-established cell line generated in the lab of Masaaki Komatsu (*Komatsu M, et al. J Cell Biol 169: 425-434*) and used in many laboratories, including our lab in Fumagalli et al 2016, Fregno et al 2018, Fregno et al 2021, Loi et al 2019, Kucinska et al 2023). We agree with the reviewer that the anti-Atg7 shows cross-reactions. We have now added a WB showing the lack of LC3 lipidation in the Atg7-KO cells exposed to nutrient deprivation (new Fig. 5D).

3: The author employed Lamp1 antibody for lysosomal staining in cells and observed a significant abundance of lysosomes in some experiments, as depicted in Figure 2C, 2D, 4I, etc. Is the phenomenon of lysosomes extensively filling the entire cell a common occurrence? Is it indicative of a normal physiological state?

There may be variations depending on the cell type used for the experiments. In the new version of the manuscript, we now present imaging data for 3 cell lines (NIH 3T3 with stable expression of NHK and ATZ (Figs. 2E-2H), MEF (Figs. 2J-2N, 4, 5, 6) and HEK293 with transient expression of ERAD clients (Figs. 3).

Minor issues :

1: Some immunofluorescence experimental data are unclear. Please request the authors to replace these with more distinct images, as seen in Figure 3B and 3E.

We hope that the quality of the new images will be considered sufficient for publication.

2: Some expressions appear to be questionable. For instance, the necessity of utilizing endolysosomes requires clarification.

For the use of endolysosomes (lysosome would be incorrect in our opinion to indicate these LAMP1/RAB7-positive degradative organelles), we now refer to the papers by Bright et al *Endolysosomes Are the Principal Intracellular Sites of Acid Hydrolase Activity* Curr Biol 2016, and the original definition by Huotari and Helenius *Endosome maturation* EMBO J 2011 (Introduction, page 2).

3: Some writing lacks precision, such as referring to FAM134B as FAM134.

Corrected, thank you

Reviewer #2 (Significance (Required)):

o General assessment:

o Advance: provide an meaningful evidence that how two degradative pathways are coordinated in maintaining ER homeostasis.

o Audience: cell biologist

o Reviewer's expertise: autophagy, vesicle trafficking, organelle biology

Reviewer #3 (Evidence, reproducibility and clarity (Required)):

In their study, Fasana and colleagues investigate protein quality control in the ER. Specifically, they test whether folding-incompetent proteins that are normally cleared by ER-associated degradation (ERAD) can also be targeted for degradation by direct vesicular transport from the ER to lysosomes in case ERAD is blocked. They show that blocking ERAD pharmacologically or genetically indeed leads to re-rerouting of an ERAD model substrate (the NHK variant of alpha-antitrypsin) to lysosomes and that this pathway requires the reticulon-like protein FAM134B, the ability of FAM134B to interact with the ubiquitin-like protein LC3 and the machinery for LC3 lipidation.

The paper is, for the most part, easy to follow. There are, however, a few minor issues and I think the authors could do more to connect their work with similar studies in the literature. Accordingly, I have some general and specific suggestions to make the manuscript more accessible for the reader.

General suggestions

1. To avoid confusion, it would be helpful to more clearly distinguish between vesicular transport to endolysosomes and autophagy. Previous work by the authors has defined a trafficking pathway from the ER to endolysosomes that appears to rely on conventional vesicle-mediated transport (Fregno et al, EMBO J 2018). This pathway delivers material from the ER lumen to the lumen of endolysosomes, which are both topologically equivalent to the extracellular space. Hence, this pathway is distinct from autophagy, which is the transport of cytoplasmic components to endolysosomes and thus the transport of material from intracellular to extracellular space. This distinction is particularly important as both vesicular ER-to-lysosome transport and autophagy of the ER involve LC3 and FAM134B, which is typically referred to as an ER-phagy receptor. To make this less confusing, it may be helpful to explain that FAM134B appears to be a multifunctional molecule that can function as a receptor for macroautophagy but also in the vesicular transport pathway studied here. In addition, it would be helpful to point out that LC3 appears to also have roles unrelated to autophagosome formation.

The reviewer is referring to the original definition of ERLAD to describe the mechanisms of clearance of ATZ polymers (Fregno et al 2018). The definition of ERLAD has now been expanded and is given, for example, in Klionsky DJ, et al (2021) Guidelines for the use and interpretation of assays for monitoring autophagy (4th edition). *Autophagy* 17: 1-382 and is explained in detail in our recent review Rudinskiy M, Molinari M (2023) ER-to-lysosome-associated degradation in a nutshell: mammalian, yeast, and plant ER-phagy as induced by misfolded proteins. *Febs Letters*: 1928-1945. Notably, the acronym ERAD for ER-associated degradation has originally been used to describe the proteasomal clearance from the ER of misfolded pro-alpha factor in a reconstituted yeast system in McCracken AA, Brodsky JL (1996) Assembly of ER-associated protein degradation in vitro: dependence on cytosol, calnexin, and ATP. *The Journal of cell biology* 132: 291-298. Only later on, the acronym has been used as an umbrella term that now covers all the pathways that control

proteasomal clearance of misfolded proteins from the ER. A short historical excursus is presented in the new introduction to better explain these issues.

It is well established that LC3 and the LC3 lipidation machinery have functions that go beyond macroautophagy (which involves double membrane autophagosomes). Micro-autophagy (or micro-ER-phagy to remain on the topic of our paper) is an example of autophagic pathway relying on ER-phagy receptor that engage LC3, on the LC3 lipidation machinery, without involving autophagosomes. This is schematically represented in the new Fig. 1B.

2. Several recent papers that appear relevant to the present study are not mentioned. In particular, Sun et al., Dev Cell 2023 (PMID: 37922908) appears worthy of discussion, as does Gonzalez et al., Nature 2023 (PMID: 37225996).

Thank you. Both papers are not directly linked to our study addressing the intervention of ERLAD pathways when ERAD activity is impaired. In particular the work of Gonzales et al describes post-translational modification of ER-phagy receptors for their activation. The Sun et al paper is not really related to the topic covered in our manuscript, but we cite it as an alternative pathway that removes ATZ from the ER (page 8).

Specific suggestions

1. Abstract: The abstract begins with "About 40% of the eukaryotic cell's proteome is synthesized ... in the ER." Similar statements can be found in many papers and purportedly reflect common knowledge. However, it is unclear where the figure of 'about 40%' comes from. It would be proper to provide a reference and demonstrate that giving such a fairly precise estimate is supported by experimental data. Alternatively, the statement could be modified to avoid being precise than is justified.

No reference is allowed in the abstract. We therefore modified the sentence as suggested by the reviewer.

2. p2: "The ER is site of gene expression in nucleated cells and ... native proteins to be delivered at their site of activity ...". There is something missing at the beginning of this sentence. Also, it should be 'delivered to their site of activity', not 'delivered at'.

Thank you

3. p2: "... by mechanistically distinct ER-phagy pathways collectively defined as ER-to-lysosome-associated degradation ERLAD." This statement suggests that all pathways subsumed under the term ERLAD are ER-phagy pathways, which I believe is misleading (see comment above on the distinction between autophagy and vesicular transport pathway).

See point 1.

4. p2: "KIF selectively ...". Please spell out KIF and explain what kind of compound it is.

Thank you, we changed to “The alkaloid kifunensine (KIF) is a cell permeable selective inhibitor of the members of the glycosyl hydrolase 47 family of α 1,2-mannosidases”

5. p3: "Notably, ERAD inhibition delays, rather than blocking degradation of ERAD clients ...". Please correct, for example: Notably, ERAD inhibition delays rather than blocks degradation of ERAD clients ...

Thank you

6. Figures 2 - 5: The number of quantified cells is given but it is not clear if experiments were done once or in biological replicates. Please indicate this in the figure legends.

N is now given for all panels in the corresponding figure legends.

7. p4: "To verify if ERAD inactivation ..." sounds odd. Less ambiguous would be 'To test whether' or 'To ask if'.

Thank you

8. p7, beginning of discussion: Please correct "delivered at" to 'delivered to'.

Thank you

Reviewer #3 (Significance (Required)):

This is a concise and convincing manuscript with a clear message. The idea that proteins that cannot be processed by ERAD can be eliminated by other means, for instance by autophagy, is not new. Similarly, the FAM134B- and LC3-dependent pathway for ER-to-lysosome transport has been described by the authors before (Fregno et al, EMBO J 2018). Furthermore, the study exclusively relies on microscopy and does not attempt to tackle new mechanistic questions. Still, this study presents a definite functional advance in our understanding of the interplay of various ER quality control pathways.

The findings presented here will be of interest mainly to molecular cell biologists working on protein quality control and organelle homeostasis. However, given the disease-relevance of misfolded proteins, and alpha-antitrypsin in particular, the impact of this study may eventually go beyond basic research and may also interest translational researchers.

Dear Prof. Molinari

Thank you for the submission of your research manuscript to our journal. We have now received the full set of referee reports that is copied below.

As you will see, all referees are very positive about the study and request only minor changes to clarify text.

Before I can accept the manuscript, I kindly ask you to format your study along the EMBO Reports style and formatting guidelines. Once this gross formatting is done and your manuscript is re-submitted, we will perform a number of quality checks on the figures, the figure legends and source data and in case further edits from the editorial side are required, we will let you know so that these minor edits can be done.

General formatting guidelines are listed below and I list here two specific requests:

1) The title may not exceed 100 characters incl. spaces. I suggest some alternatives:

FAM134B-driven ER-to-lysosome-associated degradation maintains quality control upon ERAD inhibition

FAM134B-driven ERLAD acts as failsafe mechanism upon ERAD inhibition

ERAD clients are directed to lysosome-associated degradation by FAM134B

2) Statistical analysis: please perform statistical analysis only on results obtained from at least 3 independent experiments. Please do not apply statistics if 'n' cells from one experiment were analysed, since $n = 1$ in these cases. Figure 2 e.g., reports on quantification of 'n' cells. Please ensure that these quantifications were derived from independent experiments and that the statistical analysis compares the results from these different experiments and not from the individual cells measured within one replicate, as the latter would result in pseudoreplication. You already display the data from all datapoints instead of the mean/median only, which is most welcome and complies with our journal guidelines.

General guidelines - When submitting your revised manuscript, we will require:

2) individual production quality figure files as .eps, .tif, .jpg (one file per figure).

Please download our Figure Preparation Guidelines (figure preparation pdf) from our Author Guidelines pages

<https://www.embopress.org/page/journal/14693178/authorguide> for more info on how to prepare your figures.

4) a complete author checklist, which you can download from our author guidelines (). Please insert information in the checklist that is also reflected in the manuscript. The completed author checklist will also be part of the RPF.

5) Please note that all corresponding authors are required to supply an ORCID ID for their name upon submission of a revised manuscript (). Please find instructions on how to link your ORCID ID to your account in our manuscript tracking system in our Author guidelines

()

6) We replaced Supplementary Information with Expanded View (EV) Figures and Tables that are collapsible/expandable online. A maximum of 5 EV Figures can be typeset. EV Figures should be cited as 'Figure EV1, Figure EV2' etc... in the text and their respective legends should be included in the main text after the legends of regular figures.

7) Please note that a Data Availability section at the end of Materials and Methods is now mandatory. In case you have no data

that requires deposition in a public database, please state so instead of refereeing to the database.

See also < <https://www.embopress.org/page/journal/14693178/authorguide#dataavailability>>. Please note that the Data Availability Section is restricted to new primary data that are part of this study.

8) At EMBO Press we ask authors to provide source data for the main figures. Our source data coordinator, Hannah Sonntag, has already contacted you on March 21st with a detailed list of figure panels we would need source data for. She can also provide you with helpful tips on how to upload and organize the files.

Additional information on source data and instruction on how to label the files are available .

10) Figure legends and data quantification:

- the name of the statistical test used to generate error bars and P values,
 - the number (n) of independent experiments (please specify technical or biological replicates) underlying each data point,
 - the nature of the bars and error bars (s.d., s.e.m.)
-
- If the data are obtained from n {less than or equal to} 5, show the individual data points in addition to the SD or SEM.
 - If the data are obtained from n {less than or equal to} 2, use scatter blots showing the individual data points.

11) Our journal encourages inclusion of *data citations in the reference list* to directly cite datasets that were re-used and obtained from public databases. Data citations in the article text are distinct from normal bibliographical citations and should directly link to the database records from which the data can be accessed. In the main text, data citations are formatted as follows: "Data ref: Smith et al, 2001" or "Data ref: NCBI Sequence Read Archive PRJNA342805, 2017". In the Reference list, data citations must be labeled with "[DATASET]". A data reference must provide the database name, accession number/identifiers and a resolvable link to the landing page from which the data can be accessed at the end of the reference. Further instructions are available at .

12) All Materials and Methods need to be described in the main text. We would encourage you to use 'Structured Methods', our new Methods format. According to this format, the Methods section should include a Reagents and Tools Table (listing key reagents, experimental models, software and relevant equipment and including their sources and relevant identifiers) followed by a Methods and Protocols section in which we encourage the authors to describe their methods using a step-by-step protocol format with bullet points, to facilitate the adoption of the methodologies across labs. More information on how to adhere to this format as well as downloadable templates (.doc or .xls) for the Reagents and Tools Table can be found in our author guidelines: < <https://www.embopress.org/page/journal/14693178/authorguide#manuscriptpreparation>>.

An example of a Method paper with Structured Methods can be found here: .

13) As part of the EMBO publication's Transparent Editorial Process, EMBO Reports publishes online a Review Process File to accompany accepted manuscripts. This File will be published in conjunction with your paper and will include the referee reports, your point-by-point response and all pertinent correspondence relating to the manuscript.

14) Finally, EMBO Reports papers are accompanied online by A) a short (1-2 sentences) summary of the findings and their significance, B) 2-3 bullet points highlighting key results and C) a synopsis image that is 550x300-600 pixels large (width x height) in PNG for JPG format. You can either show a model or key data in the synopsis image. Please note that the size is rather small and that text needs to be readable at the final size. Please send us this information along with the revised manuscript.

Kind regards,

Referee #1:

The authors have sufficiently addressed the issues raised with their additional data in the manuscript and rebuttal letter. A very interesting piece of work that will be important for the field and also a general audience.

Referee #2:

The authors have provided sufficient experimental evidence to address my concerns and have also answered the scientific questions raised. I believe this manuscript is suitable for publication in EMBO Reports journal.

Referee #3:

When commenting on this manuscript for Review Commons, I raised essentially one point that I felt should be improved. This was the definition of the three transport pathways from/of the ER to lysosomes that the authors refer to by the umbrella term 'ERLAD'. The explanation in the introduction is now clearer than before. However, a potential source of confusion still is that FAM134B is consistently referred to as 'the ER-phagy receptor FAM134B', which may create the misleading impression that FAM134B is only involved in autophagic pathways. To avoid this misunderstanding, the authors could explain the different functions of FAM134B in the introduction and then omit the descriptor 'ER-phagy receptor' throughout the text. This, however, is for the authors to decide.

Other than that, I believe that the manuscript is now fit for publication in EMBO Reports.

Rev_Com_number: RC-2023-02281

New_manu_number: EMBOR-2024-59101V1-T

Corr_author: Molinari

Title: FAM134B-driven ER-to-lysosome-associated degradation intervention upon pharmacologic or genetic inactivation of ER-associated degradation

All editorial and formatting issues were resolved by the authors.

Dear Prof. Molinari

Thank you for the submission of your revised manuscript. We have now performed all figure and editorial checks on your manuscript and there are a number of things that need your attention before we can proceed with official acceptance of your study:

- Author Checklist, Design - Laboratory protocol: This section specifically asks whether your study includes step-by-step protocols. Please carefully check and only choose "Materials and Methods" if such information is part of your methods section or if you refer to such a detailed method.
- Your manuscript file contained figures. We have removed these for you, but please be reminded to exclude figures from the manuscript text file when you upload the final version.
- Please reduce the number of keywords to 5. I suggest removing Endolysosome and Proteasome.
- Please move the Data Availability section before the Acknowledgments.
- The information on funding should be part of the Acknowledgments.
- Please update the 'Conflict of interest' paragraph to our new 'Disclosure and competing interests statement'. For more information see <https://www.embopress.org/page/journal/14693178/authorguide#conflictsofinterest>
- Regarding the Author Contributions, we now use CRedit to specify the contributions of each author in the journal submission system. Therefore, please remove the Author Contributions from the manuscript file and make sure that the author contributions in our manuscript tracking system are correct and up-to-date. The information you specified in the system will be automatically retrieved and typeset into the article. You can enter additional information in the free text box provided, if you wish.
- There is no callout to Figure 2, panel J in the text. Please add it wherever appropriate.
- The abstract has a rather long introduction and only a short description of the new findings. I suggest to expand the latter. Please see my suggestion pasted below my signature. I also noticed that you never define the abbreviation NHK in the manuscript.
- At final size of 550 pixels width (and 100% zoom), the text of the synopsis image is difficult to read. Please adjust the text size so that it is legible at this image size.
- Figure 6 legend mentions "2A-2C for BACE457[delta]. Densitometric quantification of the 35S-NHK band in the representative experiment in panel A is shown". Should this not rather be "of the 35S-BACE457delta band"?
- Figure 6G and O provide statistical significance based on 2 experiments and should therefore be removed. Basing statistics on "n" cells from the same experiment results easily in pseudoreplication, since these cells are not "independent entities/experiments".
- Source data need to be unzipped and uploaded as one zipped folder per figure and the comments in the readme.txt should be inserted in the SD checklist.
- Our production/data editors have asked you to clarify several points in the figure legends (see below). Please incorporate these changes in the manuscript and return the revised file with tracked changes with your final manuscript submission.
 - 1) Please note that a separate 'Data Information' section is required in the legends of figures 3b, e; 4b, d-j; 5a-c, e, g; 6d-f, h-n. (This means if certain conditions like "n" or "mean +/- SD" are shared between certain panel, they can be summarized in a "Data information: " section at the end of the legend.)
- On a different note, I would like to alert you that EMBO Press offers a new format for a video-synopsis of work published with us, which essentially is a short, author-generated film explaining the core findings in hand drawings, and, as we believe, can be very useful to increase visibility of the work. This has proven to offer a nice opportunity for exposure i.p. for the first author(s) of the study. Please see the following link for representative examples and their integration into the article web page:

https://www.embopress.org/video_synopses
<https://www.embopress.org/doi/full/10.15252/emj.2019103932>

Please let me know, should you be interested to engage in commissioning a similar video synopsis for your work. According

operation instructions are available and intuitive.

With kind regards,

Abstract suggestion:

The endoplasmic reticulum (ER) produces proteins destined to organelles of the endocytic and secretory pathways, the plasma membrane, and the extracellular space. While native proteins are transported to their intra- or extra-cellular site of activity, folding-defective polypeptides are retro-translocated across the ER membrane into the cytoplasm, poly-ubiquitylated and degraded by 26S proteasomes in a process called ER-associated degradation (ERAD). Large misfolded polypeptides, such as polymers of alpha1 antitrypsin Z (ATZ) or mutant procollagens, fail to be dislocated across the ER membrane and instead enter ER-to-lysosome-associated degradation (ERLAD) pathways. Here, we show that the ERAD clients misfolded Null Hong Kong α 1-Antitrypsin (NHK) and BACE457[delta] are delivered to LAMP1-positive endolysosomes in cells with dysfunctional ERAD. Pharmacological or genetic inhibition of ERAD components, such as the [alpha] 1,2-mannosidase EDEM1 or the OS9 ERAD lectins triggers the delivery of both clients to degradative endolysosomes dependent on the ERLAD component FAM134B and a functional autophagy machinery. Our results reveal that ERAD dysfunction is compensated by activation of FAM134B-driven ERLAD pathways that ensure efficient lysosomal clearance of orphan ERAD clients.

Rev_Com_number: RC-2023-02281

New_manu_number: EMBOR-2024-59101V2

Corr_author: Molinari

Title: ER-to-lysosome-associated degradation acts as failsafe mechanism upon ERAD dysfunction

The authors have addressed all minor editorial requests.

Prof. Maurizio Molinari
Institute for Research in Biomedicine
Protein Folding and Quality Control
Via F. Chiesa 5
Bellinzona, Ticino CH-6500
Switzerland

Dear Mauri,

I am very pleased to accept your manuscript for publication in the next available issue of EMBO reports. Thank you for your contribution to our journal.

Kind regards,

Martina

Rev_Com_number: RC-2023-02281
New_manu_number: EMBOR-2024-59101V3
Corr_author: Molinari
Title: ER-to-lysosome-associated degradation acts as failsafe mechanism upon ERAD dysfunction